# Preparation of Cotton Linters’ Aerogel-Based C/NiFe_2_O_4_ Photocatalyst for Efficient Degradation of Methylene Blue

**DOI:** 10.3390/nano12122021

**Published:** 2022-06-11

**Authors:** Chengli Ding, Huanhuan Zhao, Xiao Zhu, Xiaoling Liu

**Affiliations:** Key Laboratory of Coal Cleaning Conversion and Chemical Engineering Process, Xinjiang Uygur Autonomous Region, Xinjiang University, Urumqi 830046, China; zhaohuanhuan@stu.xju.edu.cn (H.Z.); zhuxiao@stu.xju.edu.cn (X.Z.); linglongliu@sina.com (X.L.)

**Keywords:** cellulose, carbon aerogel, NiFe_2_O_4_, photo-Fenton, magnetism

## Abstract

At present, the research focus has been aimed at the pursuit of the design and synthesis of catalysts for effective photocatalytic degradation of organic pollutants in wastewater, and further exploration of novel materials of the photodegradation catalyst. In this paper, the Sol-gel route after thermal treatment was used to produce NiFe_2_O_4_ carbon aerogel (NiFe_2_O_4_-CA) nanocomposites with cotton linter cellulose as the precursor of aerogel, by co-precipitating iron and nickel salts onto its substrate. The structure and composition of these materials were characterized by X-ray diffraction (XRD), energy dispersive spectroscopy (EDS), Raman spectra, high-resolution scanning electron microscopy (HR-SEM), high-resolution scanning electron microscope mapping (SEM-mapping), X-ray photoelectron spectroscopy (XPS) and Brunauer–Emmett–Teller (BET)’s surface area. The magnetic properties of the material were analyzed by a vibrating-sample magnetometer (VSM). Moreover, diffuse reflectance spectra (DRS), electrochemical impedance spectroscopy (EIS) and photo-luminescence spectroscopy (PL) characterized the photoelectric properties of this cellulose-aerogels-based NiFe_2_O_4_-CA. Methylene blue (MB) acted as the simulated pollutant, and the photocatalytic activity of NiFe_2_O_4_-CA nanocomposites under visible light was evaluated by adjusting H_2_O_2_ content and the pH value. The results showed that the optical absorption range of nickel ferrite was broadened by doping cellulose-aerogels-based carbon, which exerted more positive effects on photocatalytic reactions. This is because the doping of this aerogel carbon promoted a more uniform distribution of NiFe_2_O_4_ particles. Given the Methylene blue (MB) degradation reaction conformed to the first-order kinetic equation, the NiFe_2_O_4_-CA nanocomposites conducted excellent catalytic activity by maintaining almost 99% of the removal of MB (60 mg/L) within 180 min and upheld excellent stability over four consecutive cycles. This study indicated that NiFe_2_O_4_-CA nanocomposites reserved the potential as a future effective treatment of dye wastewater.

## 1. Introduction

In recent years, photocatalysis, as one of the advanced oxidation processes, has provided a promising pathway for solar energy conversion [1], which reserves broad prospects of applications in wastewater treatment, air purification, solar cells, antibacterial agents and many other fields. Current exploration of the highly active photocatalyst functioning in the visible lights has attracted remarkable attention [2,3,4]. Nickel ferrite (NiFe_2_O_4_) is one of the most interesting ferrites thanks to the dependence of its performance on particle size. NiFe_2_O_4_ nanoparticles have a considerable photo-response within the visible light region with good photochemical stability, suggesting their potential as semiconductors’ photocatalysts [5,6]. However, the single pure NiFe_2_O_4_ exhibited a low efficiency under visible light irradiation because of its low conductivity, easy aggregation, and quick recombination of photo-generated electron-hole pairs. In addition, the size, morphology and active surface area of NiFe_2_O_4_ particles all affect their photocatalytic performance. To overcome these disadvantages, researchers have paid much attention to the composites of NiFe_2_O_4_ nanoparticles with other materials [6,7,8,9]. Although ferrite is a type of photocatalytic inert material, its photocatalytic performance can be improved by combining it with an appropriate carrier. In the system of heterogeneous catalytic reaction, the function of the carrier can not only keep the dispersion of the metal particles but also plays an important role in maintaining the efficiency and stability of the catalytic system [10].

Moreover, there have been numerous merits of using carbon material as catalyst carriers [11]: (1) rich pore structure; (2) its surface chemical properties are easy to regulate; (3) conducive to the reduction of metal phase; (4) excellent acid and alkaline resistances; (5) high-temperature stability; (6) this porous carbon material can be prepared in different shapes, such as fibrous, granular and spherical; (7) its active components are easy to be recovered; (8) low cost. Based on the above merits, it can exert a positive effect on the development of heterogeneous catalysis. The normal carbonaceous materials are usually based on graphene, carbon nanofibers and nanotubes extracted from dwindling petroleum resources, which are expensive and complex to refine [12]. Therefore, biomass conversion to carbonaceous materials has received increasing attention, which is regarded as an alternative to the usual management of solid waste because it potentially replaces fossil fuels [13]. Cellulose is the main component of a plant (over 50%), which is the most abundant and widely available biomass energy in nature. Meanwhile, cellulose has been widely used in biomedical and pharmaceutical fields attributed to its good reusability, environmental friendliness, biodegradability and bio-compatibility [14]. Therefore, it can be utilized as a precursor of carbon [15]. Due to the aggregation of cellulose chains (the formation of substantial hydrogen bonds between the cellulose chains), it can hardly be dissolved in solvents but can readily be formed into gels in an anti-solvent. Subsequently, the cellulose is treated through a low-temperature, freeze-dried and carbonized technology, which leads to the generation of three-dimensional layered porous aerogels and carbon ones. Carbon aerogels are based on those with an adjustable porous structure, large specific surface area and benign electrical conductivity [16], so they reserve a broad prospect for application as adsorptive and catalytic materials.

In this paper, abundant cotton linters in Xinjiang were adopted as a carbon aerogel precursor relying on the alkaline substrate of cellulose hydrogel, so nickel ferrite nano-metal compounds were synthesized and encapsulated into cellulose-based carbon aerogels to form the cotton linters’ cellulose-based NiFe_2_O_4_ carbon aerogel (NiFe_2_O_4_-CA) nanocomposites. This route imparts magnetic properties to carbon nanoparticles, which enable them to avoid the secondary contamination caused by the leaching of NiFe_2_O_4_ nanoparticles into the solution under the heterogeneous system. The cotton linters’ cellulose-based NiFe_2_O_4_ carbon aerogel nanocomposite was cleverly constructed into a REDOX system by applying the photo-Fenton principle to increase the generation of OH free radicals and to promote the transformation of pollutants. The research of this item presents a new practical and effective method for the treatments of printing and dyeing wastewaters.

## 2. Materials and Methods

### 2.1. Materials

Cotton linters (cellulose, ≥94%, Aksu, Xinjiang, China), iron (III) nitrate nonahydrate (Fe(NO_3_)_3_·9H_2_O, 98%, Yongsheng Fine Chemicals, Tianjin, China), nickel (II) nitrate hexahydrate (Ni(NO_3_)_2_·6H_2_O, 99%, Shanpu Chemical, Shanghai, China), hydrogen peroxide (H_2_O_2_, 30%, Yongsheng Fine Chemicals, Tianjin, China), Methylene blue and urea (Bodi Chemical, Tianjin, China) were purchased and applied respectively. All the reagents used were of AR grade and were adopted as soon as they were received. The chemical composition of cotton linters is shown in Table 1.

### 2.2. Methods

#### 2.2.1. Preparation of Cotton Linters’ Cellulose/NiFe_2_O_4_ Carbon Aerogel Nanocomposites

A total of 4g of cotton linters’ cellulose was dispersed into the green solvent system of NaOH/Urea/H_2_O (mass ratio: 7:12:81), which was frozen for 12 h at −20 °C to form cellulose hydrogel. Nickel nitrate hexahydrate (Ni(NO_3_)_2_·6H_2_O) and ferric nitrate ix hydrate (Fe(NO_3_)_3_·9H_2_O) were both added to cellulose hydrogels at the molar ratio of 1:2. The load ratio of cellulose was changed by adjusting the masses of Ni(NO_3_)_2_·6H_2_O and Fe(NO_3_)_3_·9H_2_O. The above samples were heated and stirred in a water bath at 80 °C for 4 h, and then strongly stirred for 40 min. The hybrid colloids with 25%, 50%, 75% and 100% cellulose nickel (NiFe_2_O_4_) were prepared respectively, which were washed with ethanol and distilled water after they were naturally cooled to room temperature until the filter cakes became neutral. Afterwards, the filter cakes were freeze-dried at −60 °C for 24 h in a freeze-drying machine to formulate the cotton linters’ cellulose-based NiFe_2_O_4_ composite aerogels. Then, they were carbonized in a tube furnace with an N_2_ atmosphere setting the temperature at 600 °C for 2 h, until cellulose/NiFe_2_O_4_-CA samples were eventually gained. The optimum calcination temperature for obtaining the optimal photocatalytic activity of NiFe_2_O_4_-CA under the conditions is 600 °C. The formation route of NiFe_2_O_4_-CA is shown in Figure 1.

#### 2.2.2. Characterization Methods

X-ray diffraction (XRD) patterns were recorded for phase analysis. The morphological structures of the prepared NiFe_2_O_4_-CA NPs were studied by a high-resolution transmission electron microscope (TEM) (Peabody, MA, USA). The crystallinity of the materials was evaluated by analysis of high-resolution XRD (D8 Discovery, Bruker, Berlin, Germany) in the Bragg (reflection) geometry with a pure Cu kα1 radiation (wavelength; λ; 1.54056 Å). The Raman spectrum was obtained through a Raman spectroscopy system (Lab Ram HR 800, Horiba JY, Kuoto, Japan) with an Ar^+^ laser (514 nm wavelength) as the excitation source. The elemental analysis of the NiFe_2_O_4_-CA was conducted by XPS (ESCALAB, 250Xi, ThermoFisher Scientific, Waltham, MA, USA).

#### 2.2.3. Photocatalytic Performance Test

In this investigation, an MB solution was employed to simulate pollutants, while a xenon lamp of 120 W was used as the light source with applied mechanical stirring. The photodegradation performance was examined by a UV-visible spectrophotometer. Firstly, after the adsorption equilibrium curve of cellulose/NiFe_2_O_4_-CA in the pollutant solution was studied, the light source was turned on to explore its photodegradation effect on the contaminant. The change of absorbance value reflected the degradation effect. The degradation efficiency η = (*C_0_* − *C*)/*C_0_*, *C_0_* is the initial concentration value of the MB solution, and *C* represents this value, after a certain moment.

## 3. Results and Discussions

### 3.1. Composition Characterizations

XRD patterns of the as—synthesized cellulose NiFe_2_O_4_-CA are shown in Figure 1a. With the increase in temperature, the XRD peak of the NiFe_2_0_4_ sample is obvious, sharp, and the most outstanding at 600 °C. All the diffraction peaks in Figure 1a could be well indexed with the cubic spinel NiFe_2_O_4_ having JCPDS card No. 86-2267 [17]. XRD patterns of NiFe_2_O_4_ at 2ϴ = 18.4°, 30.3°, 35.7°, 37.3°, 43.4°, 53.8°, 57.4°, 62.9° and 74.6° individually correspond to (111), (220), (311), (222), (400), (422), (511), (440) and (533) crystal planes. The average grain size of NiFe_2_O_4_ is 2.5 nm according to the Debye–Scherrer formula. However, no typical diffraction peak of carbon aerogel is observed in the XRD pattern for NiFe_2_O_4_-CA. This may be ascribed to the fact that crystal growth of NiFe_2_O_4_ between the interlayers of carbon aerogels broke the regular layer stacking, leading to the exfoliation of carbon aerogels and the disappearance of the diffraction peak [12,18,19].

The existence of carbon was verified by EDS spectra of the cotton linters’ cellulose-based NiFe_2_O_4_ carbon aerogel nanocomposites in Figure 1b. As exhibited in Figure 1c, the I_D_/I_G_ ratio increased slightly from 0.85 to 0.89, indicating that the degree of graphitization hardly changed at this carbonization temperature of 600 °C.

The surface elementary composition and chemical state of cellulose/NiFe_2_O_4_-CA were further identified by XPS experiment. In the survey spectrum of XPS, the binding energies of 284.9, 536.2, 709.8 and 856.3 eV correspond to the rated ones of C, O, Fe and Ni in Figure 1d, respectively, which proved that the composite materials contained the elements of C, O, Fe and Ni. The C1s high-resolution spectrum (Figure 1e) was fitted at 284.8 eV and 286.8 eV as two different peaks corresponding to those of C-C ring and epoxy group (C-O) in the cellulose structure [20]. The characteristic peak of O1s appeared at the binding energy of 532.5 eV due to O^2−^ in NiFe_2_O_4_ metal oxide with all probability in Figure 1f [21]. There are two typical binding energy peaks of the high-resolution Fe2p decomposition spectrum in Figure 1g, namely 712.0 and 725.2 eV, which are geared to the binding energies of Fe2p_3/2_ and Fe2p_1/2_ split orbitals, respectively. The difference between them indicated that Fe in cotton linter cellulose-based NiFe_2_O_4_ carbon aerogel nanocomposite existed in the form of Fe^3+^, which was basically consistent with the results reported in the literature [22,23]. Furthermore, high-resolution spectral lines of Ni2p formed two main peaks as shown in Figure 1h, Ni2p orbit was split into two orbitals of Ni2p_3/2_ and Ni2p_1/2_ with binding energies of 854.5 and 874.3 eV, respectively, which demonstrated the existence of Ni (Ⅱ) Ni2p [19,24].

### 3.2. Morphology Characterization

The morphology of the samples was characterized by a high-resolution scanning electron microscope (HR-SEM). It observed that cellulose aerogels exhibited a three-dimensional network structure with porosity (Figure 2a). When it was carbonized at 600 °C, the formed carbon aerogels still revealed a porous frame (Figure 2b). This fact indicated the presence of the C-C skeleton, which could provide sites for the NiFe_2_O_4_ loading after carbonization. In this study, NiFe_2_O_4_ was manufactured by co-precipitation with a relatively regular spherical structure as shown in Figure 2c. The cotton linters’ cellulose-based NiFe_2_O_4_ hybrid aerogels were carbonized to generate their carbon aerogel nanocomposites, as shown in Figure 2d–f. From the figures, it could be seen that the spherical particles were loaded onto the three-dimensional framework structure. After that, the element distribution of cellulose/NiFe_2_O_4_-CA was analyzed by EDS-mapping characterization (Figure 2g). The density of Fe elements is higher than that of Ni elements, corresponding to the molar ratio and mass ratio of NiFe_2_O_4_. Oxygen was relatively distributed widely due to the highest quantity of oxygen atoms within the NiFe_2_O_4_ molecules.

### 3.3. Analysis of Vibrating Sample Magnetometer (VSM)

The magnetic field intensity of NiFe_2_O_4_, cellulose/NiFe_2_O_4_-CA and cellulose/NiFe_2_O_4_ were separately characterized by VSM within the magnetic field strength range of ± 1.5 T at 300 K. It could be seen from Figure 3a that the hysteresis loops of the three samples are S-type curves and all symmetric about the origin; hence, the conclusion can be drawn that the NiFe_2_O_4_, cellulose/NiFe_2_O_4_-CA, and cellulose/NiFe_2_O_4_ are all super-paramagnetic materials with coerced and remanent magnetization of zero. However, the magnetism was enhanced significantly for cellulose/NiFe_2_O_4_-CA, whose saturation magnetization was 68.95 emu/g contrasted to the 29.23 emu/g of the single NiFe_2_O_4_. This fact can be interpreted as that the surface of nickel ferrite is covered by thick aerogels, resulting in the weakening magnetism. While cotton linters’ cellulose/NiFe_2_O_4_ are treated via carbonization, cellulose pyrolysis-made NiFe_2_O_4_ loaded on the skeleton structure no longer thickened as before. In the meantime, carbon aerogel as the substrate affects NiFe_2_O_4_ dispersion so that NiFe_2_O_4_ nanoparticles are no longer agglomerated; therefore, cellulose/NiFe_2_O_4_-CA nanocomposites present a relatively ideal magnetism. Based on this particular attribute, it is convenient to recycle and reuse this specific material in wasted water.

### 3.4. The Analyses of N_2_ Adsorption/Desorption Isotherms

Figure 3b–d exhibited the individual isothermal curves of N_2_ desorptions of NiFe_2_O_4_, cellulose/NiFe_2_O_4_-CA and cellulose/NiFe_2_O_4_, which belonged to the type IV isothermal ones with H3 hysteresis loop. According to the Brunauer–Emmett–Teller (BET) model, the specific surface area (SSA) data listed in Table 2 display that the cellulose/50%NiFe_2_O_4_-CA—600 °C had a considerable SSA of 106.3 m^2^/g, while cellulose/50%NiFe_2_O_4_ had an SSA of only 9.2 m^2^/g. This phenomenon may be rooted in NiFe_2_O_4_ loads on the aerogel being tightly wrapped, causing the loss of their SSA and occupying the surface of the aerogel. However, as cellulose carbon aerogel nanocomposites were produced at high temperatures, other ingredients of cellulose may be removed to such extent that NiFe_2_O_4_ attached to the skeleton of CC and formed into a porous carbon structure as the skeleton of carbon aerogel nanocomposite materials. From the data of mean pore size, it can be seen that cellulose/NiFe_2_O_4_-CAs are mesoporous.

### 3.5. Photoelectric Property Analysis

The process of photocatalytic degradation should pay attention to the separation efficiency of the photoelectron-hole. There are two basic characterization methods to describe: the electrochemical impedance spectrum (EIS) and the steady-state/transient fluorescence spectra (FLS) in Figure 4a,b. Generally, the lower the photoluminescence (PL) intensity is, the poorer the electron-hole recombination will be, and the more beneficial to generating more excellent photodegradation. Moreover, NiFe_2_O_4_ and cellulose/NiFe_2_O_4_-CA were motivated under the excitation wavelength of 325 nm, as shown in Figure 4a. The PL intensity of cotton linter cellulose/NiFe_2_O_4_-CA was remarkably weaker than that of NiFe_2_O_4_. It attested that PL was quenched with the introduction of cellulose carbon aerogels. That is because the photogenerated electrons were effectively transferred from NiFe_2_O_4_ to the substrate of cellulose carbon aerogel [25]. The relative radius of the arcs in the Nyquist plots corresponds to the magnitude of the charge transfer resistance and the separation efficiency of the photogenerated electron-hole pairs. The smaller the radius of the impedance spectrum, the higher efficiency of the electron-hole separation and resultantly, the more excellent the photocatalytic performance. Figure 4b illustrates that the radius of the impedance spectrum of cellulose/NiFe_2_O_4_-CA is smaller than that of NiFe_2_O_4_. It can be explained that the carbon aerogel plays the role of electron transfer for cellulose/NiFe_2_O_4_-CA, which improves the separation efficiency of the NiFe_2_O_4_ electron-hole pair, leading to the faster transfer of interface charge. To sum up, cellulose carbon aerogel can capture electrons, reducing the electron-hole pair recombination efficiency when NiFe_2_O_4_ nanoparticles are loaded.

The optical properties of the materials were separately measured by UV-visible diffuse reflectance spectroscopy (DRS) and are exhibited in Figure 4c,d. The NiFe_2_O_4_, cellulose/NiFe_2_O_4_-CA and cellulose/NiFe_2_O_4_ can absorb lights within the visible range in Figure 4c, which demonstrated that the cellulose/NiFe_2_O_4_-CA suffered red shift, which obviously broadened the absorption range of visible light wavelength and provided strong ability of responses to both ultraviolet and visible wavelength. The bandgap width of the material is estimated by the Tauc plot method in hν against (αhν)^2^. The intersection point of the straight line in Figure 4d extrapolated to the horizontal axis is the bandgap width Eg. The Egs of NiFe_2_O_4_ and cellulose/NiFe_2_O_4_-CA are identified as 2.3 and 2.0 eV respectively, which are almost consistent with the results reported in the literature [21,26]. It indicates that the introduction of cellulose carbon aerogels widened the range of light absorptions, which may exert more positive effects on photocatalytic reactions.

### 3.6. Photocatalytic Performance and Mechanism

The performance of cellulose-based composite carbon aerogels with different NiFe_2_O_4_ loads was measured at room temperature. To eliminate the interference of adsorption in degradation, adsorption experiments were conducted under dark conditions, as shown in Figure 5a. The light source is turned on when the adsorption equilibrium is reached (Figure 5b). It is obvious that cellulose/NiFe_2_O_4_-CA exhibits better adsorption-degradation performance. As the loading ratio surpassed 50%, the performance of cellulose/NiFe_2_O_4_-CA declined, which resulted from the agglomeration of NiFe_2_O_4_ nanoparticles, bringing about the decrease in SSA of cellulose/NiFe_2_O_4_-CA, which affected the photocatalytic activity.

Figure 5c reveals the influences of calcination at different temperatures on the photodegradation of cotton linter cellulose/NiFe_2_O_4_-CA. It can be verified that 600 °C is the optimum calcining temperature for carbonization. When the calcination temperature turns to 400 °C, the carbonization is likely to be incomplete. The adsorption–degradation curves manifested that the cellulose/NiFe_2_O_4_-CA reserved a certain ability of response to light, but the photodegradation rate fell to only about 10% within 3 h. Furthermore, the addition of H_2_O_2_ exerted a positive effect on facilitating photocatalysis, because when 5 mmol H_2_O_2_ was added to the pollutant solution with MB concentration of 20 mg/L, its removal was able to reach 30% within 3 h, as shown in Figure 5d. It can be seen as non-negligible for photodegradation to identify the addition dosage of H_2_O_2_. However, from the economic perspective and degradation effect, although the photodegradation efficiency has been improved when the dosage of H_2_O_2_ is more than 20 mmol, the decolorization ratio remains very inconspicuous, so it is not advisable to blindly raise the degradation rate and gain a little degradation effect at the expense of H_2_O_2_; therefore, 20 mmol H_2_O_2_ is considered more appropriate. It can be seen from Figure 5e that the photodegradation of MB solution by cellulose/NiFe_2_O_4_-CA conforms to the first-order kinetic equation ln(*C/C_0_*) = −*k*t. Table 3 lists the reaction rate constants with different dosages of H_2_O_2_. There existed 20 mmol H_2_O_2_, and the photodegradation of MB solution of 20 mg/L can achieve 99% within 180 min. The full spectrum of MB UV-visible absorption wavelength in Figure 5f corresponds to the degradation in Figure 5d. With the increase of degradation time, the absorbance value of MB solution decreases. When the degradation lasts for 180 min, almost no absorption peak appears on the absorption spectrum.

The effect of a heterogeneous photocatalytic reaction is not only affected by the semiconductor-loading mass of NiFe_2_O_4_ and the concentrations of H_2_O_2_, but also impacted by the pH value. Therefore, it is necessary to set different pH values to investigate the corresponding changes of photocatalytic degradation. It can be seen from Figure 6a that an alkaline condition is more suitable for photodegradation. For the 20 mg/L MB solution under the alkaline condition of pH = 11, the cotton linter cellulose/NiFe_2_O_4_-CA can eliminate it within 1 h. The photodegradation kinetics curves of different pH were fitted, the results of which are exhibited in Figure 6b. The reaction rate constants at pH = 3, 5, 7, 9 and 11 were 0.013, 0.018, 0.021, 0.035 and 0.088 min^−1^, respectively, by linear fitting. Among them, the acidic environment was found to inhibit the degradation. The excellent photocatalytic effect of cellulose/NiFe_2_O_4_-CA is under the joint action of light, H_2_O_2_ and the proper pH value. The efficiency of a single condition or the unloading material turns out to be far weaker than that under this joint action and cellulose load NiFe_2_O_4_ from Figure 6c.

Under the same condition, the degradation capacity of cellulose/NiFe_2_O_4_-CA descended with the increase of pollutant concentration, as well as the degradation rate constant decreased from 0.058 min^−1^ to 0.027 min^−1^, as shown in Figure 7a. This phenomenon can be explained as follows: (1) The higher the pollutant concentration, the weaker the ability of light to penetrate through solution, and the fewer photons are involved in the catalytic oxidation reaction; (2) The higher the pollutant concentration is, the more MB particles are adsorbed on the surface of cellulose/NiFe_2_O_4_-CA, and the lower quantity of the effective photocatalytic active sites. As a consequence, the electron and hole pairs produced by light per unit of time are also reduced. (3) The higher the pollutant concentration, the more intermediate products produced by the reaction may be re-adsorbed on the surface of the catalyst before the complete decomposition. Nevertheless, the removal efficiency of 50 mg cellulose/50%NiFe_2_O_4_-CA-600 °C reached 99% for 60 mg/L within 180 min. According to Figure 7b, it is not difficult to find that the number of catalysts also played a crucial role in photodegradation. The efficiency of degradation showed a positive correlation with the dose of cellulose/NiFe_2_O_4_-CA, and the reaction rate constant rose from 0.012 min^−1^ to 0.037 min^−1^ when the concentration of catalysis increased from 20 mg/L to 60 mg/L.

As the growth of the catalytic dose leads to the increase in the active site of degradation reaction, it becomes favorable to the generation of ·OH radicals. Cellulose carbon aerogel plays an adsorption agent in removing MB attributed to its large SSA of the three-dimensional reticular skeleton under dark conditions. When valence electrons of composites were excited by illumination, the cellulose/NiFe_2_O_4_-CA, the light and H_2_O_2_ jointly constructed a photo-Fenton system. NiFe_2_O_4_ generated a photoelectron-hole as shown in Equation (1), and photo-electrons migrated to the surface of cellulose carbon aerogel (Equation (2)) so that NiFe_2_O_4_ nanoparticles and carbon aerogels worked synergistically to promote the separation of a photon-generated carrier. Thus, the recombination of electrons and holes was restricted. Valence band (VB) electron excitation transfers to the conduction band (CB), leaving holes in the valence band of the semiconductor, which can oxidize the donor molecule and react with the adsorbed water molecule (Equation (5) to produce hydroxyl radicals (·OH). Meanwhile, the hole (h^+^) generated by the semiconductor NiFe_2_O_4_ is subsequently captured by OH^−^, also producing ·OH radicals, as expressed by Equation (6) [27]. ·OH radicals have a strong oxidation ability, so they are more suitable for degradation under alkaline conditions. A portion of electrons at CB reacts with dissolved oxygen to produce super-oxide ion (O^2−^). The parts captured by cellulose carbon aerogel contributed to the reduction of Fe (III) for Fe (II), which reacted with H_2_O_2_ to generate both Fe (III) and ·OH radicals [28,29]. The above processes consist of the REDOX system with the cellulose carbon aerogel’s large SSA providing active sites and facilitating electron transfers. ·OH radicals act on MB and convert it into MB molecules firstly, then degrades the macro-molecules into CO_2_ and H_2_O through the above REDOX reaction [30].
(1)$${\mathrm{NiFe}}_{2}O_{4}\overset{hv}{\to }{{\mathrm{NiFe}}_{2}O_{4}\;(e}^{-}+h^{+})$$
(2)$${\mathrm{NiFe}}_{2}O_{4}(e^{-})+\mathrm{Cellulosed}\;\mathrm{carbon}\;\mathrm{aerogel}\overset{}{\to }{\mathrm{NiFe}}_{2}O_{4}+{\mathrm{Cellulosed}\;\mathrm{carbon}\;\mathrm{aerogel}\;(e}^{-})$$
(3)$${\mathrm{Fe}}^{3+}+{\mathrm{Cellulosed}\;\mathrm{carbon}\;\mathrm{aerogel}\;(e}^{-})\overset{}{\to }{\mathrm{Fe}}^{2+}+\mathrm{Cellulosed}\;\mathrm{carbon}\;\mathrm{aerogel}$$
(4)$${\mathrm{Fe}}^{2+}+H_{2}O_{2}+e^{-}\overset{}{\to }{\mathrm{Fe}}^{3+}+\cdot \mathrm{OH}$$
(5)$$H_{2}O+h^{+}\overset{}{\to }\cdot \mathrm{OH}$$
(6)$${\mathrm{OH}}^{-}+h^{+}\overset{}{\to }\cdot \mathrm{OH}$$

### 3.7. Stability Test Experiment

To detect the stability and reusability of cellulose/NiFe_2_O_4_-CA in a circular fashion, it was recovered according to the super-paramagnetic properties of materials before the samples being washed and recovered with ethanol repeatedly. The degradation efficiency of MB fell slightly with the increase of cycles in Figure 7c. This result is attributed to the fact that the MB molecules after the previous cycle were adsorbed on the surface of the photocatalyst and the dosage of catalysts was lost in the process of the cycle, resulting in the decline of the overall effect of adsorption–photodegradation of the material. Figure 7d manifested the degradation efficiency of the four cycles experiment; the result showed that with the increase of material cycles, pollutants removal efficiency slightly decreased, but the removal efficiency of pollutants could still reach 80%. The XRD characterization of circulating utilization five times revealed that the material characteristics of diffraction peak position (Figure 1a) did not change, which could deduce that cellulose/NiFe_2_O_4_-CA in the process of circulation might reserve the stability so that it is able to be used repeatedly.

## 4. Conclusions and Outlooks

The network-structured NiFe_2_O_4_ nanocomposites based on cotton linters’ cellulose aerogels were synthesized by the co-precipitation method, showing a high crystallinity and a 3D net-like structure, which boosted the dispersion of NiFe_2_O_4_ simultaneously, while the presence of NiFe_2_O_4_ endowed cotton linter cellulose’s carbon aerogels with magnetism. The SSA of the single NiFe_2_O_4_ was 25.5 m^2^/g, which was enlarged by four times (106.3 m^2^/g) after it was supported on the surface of carbon aerogels of cotton linter cellulose. The integration of NiFe_2_O_4_ and cellulose caused redshift for the absorption of light, which broadened the absorption range of visible wavelengths and raised the absorption efficiency of light. In addition, the synergistic effect of NiFe_2_O_4_ nanoparticles and carbon aerogel effectively suppressed photoelectron-hole pairs recombination and improved quantum yields. In the advanced oxidation system, the photodegradation reaction follows the first-order kinetic equation, and the presence of an H_2_O_2_ and alkaline environment are both instrumental for the heterogeneous system to generate more ·OH radicals and accelerate the degradation of contaminants. The cycling experiments demonstrated that the materials were stable in water and not degraded in the photo-Fenton reaction. Cotton linter cellulose and NiFe_2_O_4_ are relatively more accessible but abandoned seriously in nature. It will be promising to provide this methodology for optimizing environmental governance and addressing energy dilemmas by introducing cellulose into photocatalysis. The simple design of this material will impart cellulose/NiFe_2_O_4_-CA the properties of organic matters and inorganic materials, along with the toughness of organic aerogels and rigidness of inorganic aerogels. In this study, a new strategy was proposed to prepare a promising functional material for the degradation of organic pollutants in wastewater for photo-Fenton catalytic reactions.

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
