# Peer review of "Preparation of Cotton Linters’ Aerogel-Based C/NiFe2O4 Photocatalyst for Efficient Degradation of Methylene Blue"

_nanomaterials, 2022, doi:10.3390/nano12122021_

Round 1

Author Response

Dear reviewers:

Greetings!

We apologize for the delay in answering your question for several days due to the epidemic. appreciate that you gave us a chance of major revision to improve our manuscript to a level suitable for publication in《nanomaterials》. Attached are detailed point-by-point responses and revised manuscripts.

Kind regards,

Chengli Ding

Reviewer 2 Report

Review of paper ‘Effect of cotton linter cellulose aerogel-based carbon doping on the structural, morphological, and photocatalyst dye wastewater properties of NiFe2O4’ prepared Chengli Ding, Huanhuan Zhao, Xiao Zhu and Xiaoling Liu.

The manuscript Nanomaterials-1732921 is focused on the synthesis and characterization of photocatalytic degradation catalysts for organic pollutants based on cellulose. The possibility of their use for the removal of methylene blue is discussed. In my opinion, the paper is worthy of being published in Nanomaterials. However, it needs a minor correction. Detailed concerns are presented below.

  1. In my opinion, the title does not accurately refer to the work and the results presented. If, according to the title, the effect of cotton linter cellulose is studied, the results should be compared to the results of materials that do not contain cotton. I recommend modifying the subject so that it clearly indicates the nature of the work carried out.
  2. The work requires linguistic correction. There are numerous errors in the text. For example:
  3. Abstract: The results showed that the optical absorption range of nickel ferrite was broadened by doping cellulose aerogels-based carbon, which has more positive effects on photocatalytic reactions. (…). This study indicated that NiFe2O4-CA nanocomposites have the potential to be used in the future for the treatment of dye wastewater.
  4. Introduction: In recent years, photocatalysis, as one of the advanced oxidation processes, has provided a promising path for solar energy conversion. (…) The spinel-type bimetal oxide NiFe2O4 nanoparticles have attracted extensive attention (…) In the previous research, there were almost rare reports on cotton linter cellulose-based NiFe2O4 carbon aerogel nanocomposites.
  5. The text requires some editorial corrections. For example, when specifying a mass unit, a space must be maintained between the number and the unit, see Section 2.2.2: ‘Disperse 4g m cotton’, ‘for 4h,’
  6. The quality of the schema and figures should be improved. Currently the figures are blurred, and particular legends are illegible.
  7. In section 3.5 author wrote that: ‘It can be determined that carbonization at 600 °C is the optimum calcining temperature.’ This sentence explains that most of the results presented above in the manuscript are only for this one calcination temperature. However, this explanation should have been included earlier, so that the reader would immediately know why there are no results for other temperatures.
  8. In the manuscript there is no comparison of the results obtained with those available in the literature. The dye under investigation is commonly used in studies on the removal of dyes from wastewater, so it is possible to indicate to what extent the solution proposed by the authors differs from those already in use. Only such a comparison will give credence to the fact that the solution proposed is novel and that the research direction adopted is promising and worthy of investigation. On the basis of this analysis, I propose to identify the advantages and disadvantages of the solution proposed by the authors.

Author Response

(The authors gave the same response as above.)

Reviewer 3 Report

The manuscript “Effect of cotton linter cellulose aerogel-based carbon doping on the structural, morphological, and photocatalyst dye wastewater properties of NiFe2O4” describes the preparation and characterization of some NiFe2O4-CA nanocomposites, as well as their testing as photocatalysts in the photodegradation of methylene blue pollutant. Although the subject is interesting, the poor English, numerous spelling errors, and a rather confusing presentation of the material make reading of manuscript difficult.

My observations are:

  • In the introduction, the authors should highlight the article novelty, and to explain the meaning of the sentence ”there was almost rare report on” in the last paragraph on page 2. If in the literature have been already reported similar systems, these should be cited in the text.
  • DRS = diffuse reflectance spectra
  • In the “Reagents and methods” section, major clarifications are necessary. First of all, the describing of prepared samples and their labeling is confusing and difficult to follow, and therefore the inclusion of a Table for a better identification of the samples could be helpful.
  • ferric nitrate ix hydrate should be renamed as Ferric nitrate nonahydrate
  • In the characterization of the samples, the authors refer to NiFe2O4 nanoparticles, but their characterization is not given in the text.
  • If the prepared samples displayed magnetic properties, how the authors performed the photocatalytic experiments under magnetic stirring?
  • The presentation of the physical measurements and the equipment used is missing from the manuscript.
  • In the Figures 1(a), 3(a), 4 and 5, the font is too low and very difficult to read.
  • On page 4, the explanation of Scherrer formula is unclear. The Scherrer equation is utilized to determine the crystallite size, not the “partical diameter”. In fact, the entire text in the XRD data description is confusing and needs to be clarified.
  • In the assessment of XPS spectra, a peak is attributed to epoxy group in the cellulose structure. What is the source of these epoxy groups?
  • The presentation of the prepared samples is not constant, for example in the investigation of the photocatalytic performance are included some samples that have not been presented previously. An explanation about the sample selection criteria should be included in the text. If the carbonization is incomplete at 400°C, thermogravimetric analyses are mandatory in the characterization of the samples.
  • In the adsorption process, more than 50 % of MB is adsorbed in the catalyst, before the irradiation. What happens to this amount of pollutant after the after the irradiation process is complete?
  • In Fig. 5i, I could not distinguish the conditions proposed for the photocatalysis.
  • In Fig. 7a, why C/C0 for 20 mg/L at time 0 is about 0.4, and for the other two concentrations of catalyst starts form 1.0?
  • Why the authors discuss in the text Fig. 7 prior Fig. 6?

In my opinion, before publication, this manuscript requires a major systematization of the results and a clarification of the text.

Author Response

(The authors gave the same response as above.)

Round 2

Reviewer 1 Report

1- The authors should consider without statistical analysis, the provided data has low reliability. Do replication for the experiments and provide the error bars for all the graphs.

2- Some references seem too much and not necessary for simple facts. For example, the authors cited 6 articles for this sentence: "To overcome these disadvantages, researchers have paid much attention to the composites of NiFe2O4 nanoparticles and with other materials[6-11]." another example is "Therefore, it can be utilized as a precursor of carbon [17, 18]". Furthermore, it is highly recommended to use the most recent publications related to the topic (last 5 years). It is not recommended to replace old references (e.g., ref. 9 and 32)

Author Response

Dear reviewers:

According to your comments, we have revised the manuscript extensively. If there are any other modifications we could make, we would like very much to modify them and we really appreciate your help. We hope that our manuscript could be considered for publication in “nanomaterials”. Thank you very much for your help.

Reviewer 3 Report

The authors have revised the manuscript “Effect of cotton linter cellulose aerogel-based carbon doping on the structural, morphological, and photocatalyst dye wastewater properties of NiFe2O4”, but I have still some observations:

-          In Figure 1 (a), the font of the text is too low and the figure is difficult to view.

-          In Figure 5, you have enlarged the picture, but Fig. 5 (g, h and i) are missing from the figure. Please check.

-          In Fig. 6, you have included the graphics from Fig. 5 (g, h and i, previous version of the manuscript) but the legend is from Fig 7 (also from the previous version of the manuscript).

Because the manuscript was submitted with track changes, it was very difficult for me to follow the scientific changes, and the authors must thoroughly verify the concordance between the presented text, the figures included in the manuscript and the captions (legend) of the figures.

Author Response

(The authors gave the same response as above.)
